# "I wanna live and not think about the future" what place for advance care planning for people living with severe multiple sclerosis and their families? A qualitative study

Jonathan Koffman[1,2]*, Clarissa Penfold[1], Laura Cottrell[3], Bobbie Farsides[4], Catherine J. Evans[2], Rachel Burman[5], Richard Nicholas[6], Stephen Ashford[2,7], Eli Silber[5]

1 Hull York Medical School, Wolfson Palliative Care Research Centre, Hull, United Kingdom, 2 King's College London, Cicely Saunders Institute, London, United Kingdom, 3 Faculty of Nursing, Langara College, Vancouver, Canada, 4 Brighton and Sussex Medical School, Brighton, United Kingdom, 5 King's College Hospital NHS Foundation Trust, London, United Kingdom, 6 United Kingdom Multiple Sclerosis Tissue Bank, Burlington Danes, Imperial College London, London, United Kingdom, 7 Regional Hyper-Acute Rehabilitation Unit, Northwick Park Hospital, North West University, Harrow, United Kingdom

* Jonathan.koffman@hyms.ac.uk

⊘ OPEN ACCESS

**Data Availability Statement:** All relevant data are within the paper and its Supporting Information files.

## Abstract

### Background

Little is known about how people with multiple sclerosis (MS) and their families comprehend advance care planning (ACP) and its relevance in their lives.

### Aim

To explore under what situations, with whom, how, and why do people with MS and their families engage in ACP.

### Methods

We conducted a qualitative study comprising interviews with people living with MS and their families followed by an ethical discussion group with five health professionals representing specialties working with people affected by MS and their families. Twenty-seven people with MS and 17 family members were interviewed between June 2019 and March 2020. Interviews and the ethical discussion group were audio-recorded and transcribed verbatim. Data were analysed using the framework approach.

### Results

Participants' narratives focused on three major themes: (i) planning for an uncertain future; (ii) perceived obstacles to engaging in ACP that included uncertainty concerning MS disease progression, negative previous experiences of ACP discussions and prioritising symptom management over future planning; (iii) Preferences for engagement in ACP included a trusting relationship with a health professional and that information then be shared across services. Health professionals' accounts from the ethical discussion group departed from

**Funding:** This study was generously funded by the MS Society [Grant code 93]. Catherine Evans is funded by a Health Education England/ NIHR Senior Clinical Lectureship (ICA-SCL-2015-01-001). The views expressed in this publication are those of the authors and not necessarily those of the NHS, the National Institute of Health Research or the Department of Health and Social Care.

**Competing interests:** The authors have declared that no competing interests exist.

viewing ACP as a formal document to that of an ongoing process of seeking preferences and values. They voiced similar concerns to people with MS about uncertainty and when to initiate ACP-related discussions. Some shared concerns of their lack of confidence when having these discussions.

## Conclusion

These findings support the need for a whole system strategic approach where information about the potential benefits of ACP in all its forms can be shared with people with MS. Moreover, they highlight the need for health professionals to be skilled and trained in engaging in ACP discussions and where information is contemporaneously and seamlessly shared across services.

## Introduction

Multiple sclerosis (MS) is an inflammatory disease of the brain and cord resulting in degeneration that is mostly diagnosed among people between the ages of 20 and 40 years [1]. Globally, an estimated 2.8 million people have MS [2]. People with MS represent a heterogeneous population and their needs vary according to their disability. Although some people with MS have little disability, 15% have a primary progressive course and of those with relapsing-remitting disease, at least half develop secondary progression after 10–15 years [3]. People with MS have a prolonged disease course, often lasting many decades [4]. Many spend a prolonged period 'progressively dwindling' with considerable distress associated with decline [3] punctuated by situations of clinical uncertainty and superimposed illness which can be associated with a variety of coping responses [5]. In some instances, MS can lead to death due to associated complications associated with the condition [6–8]. Approximately 40–70% of people with MS develop cognitive impairment and its effects can be profound [9–11]. Some experience a reduced ability to make decisions that affect every day functioning [12]. These complex problems not only impact people with MS directly but also family members [13, 14].

### Advance care planning and multiple sclerosis

Advance care planning is a voluntary process that supports adults to consider and share their values, goals and preferences regarding future care so that if they lose the capacity to make informed decisions for themselves health professionals and their families can provide care consistent with their wishes [15]. ACP is typically bound by a country's respective legal framework for decision making about care and treatments for adults lacking capacity. In the United Kingdom (UK), ACP is regulated by the Mental Capacity Act in England and Wales, with similar provisions by common law in Scotland and Northern Ireland [16]. Planning future care and treatment in the UK can also comprise several other legally binding processes that include Lasting Power of Attorney (LPA), a voluntary authority given by an individual to another 'decision maker' regarding either their 'property and affairs', or their 'health and welfare' that can include care and treatment. In England Wales, Health and Welfare LPAs are made when an individual has mental capacity, officially registered with the Office of the Public Guardian and only activated when that individual is unable to make decisions for themselves [17]. Under certain circumstances, individuals can also make decisions to decline but not demand

treatment. This is referred to as an Advance Decision to Refuse Treatment or ADRT referred to as an advanced directive in other countries [18–20].

Despite the scepticism of its value [21, 22], an intrinsic logic of ACP continues to drive palliative care research. Potential benefits can include providing important opportunities for discussion of diagnosis and prognosis so care and treatment are aligned with individuals' preferences, improving symptom discussions, treatment adherence and reducing misunderstandings and conflict between medical staff and families [23]. ACP may also reduce inappropriate emergency hospital admissions, lead to fewer interventions of limited or futile clinical value, lead to earlier access to palliative care, result in fewer hospital deaths and may increase rates of hospice admission or appropriate care at home [8, 24, 25]. ACP is thought to help families prepare for the death of a loved one, to resolve family conflict and lead to better bereavement outcomes [26]. Although primarily concerned with improving the appropriateness and quality of care, ACP may contribute to controlling important health spending and making more appropriate and considered use of scarce resources in end of life [27]. The COVID-19 pandemic, leading to an acute risk of deterioration and dying for many, prompted many health professionals to engage in ACP discussions with 'at risk' individuals [28–31].

Little research has examined insights about ACP among people with MS [32]. Paradoxically, there is a growing interest among individuals with MS who want to talk about their future with health professionals, but this rarely happens [33–35]. The reasons postulated for this are complex and include uncertainty inherent with MS due to its unpredictable trajectory. This makes it challenging to identify those approaching a point where mental capacity is becoming compromised and/or where life may be limited [36–40]. Additionally, health professionals may fail to initiate discussions, possibly due to their reticence to discuss disease progression and death and managing their own emotions during difficult conversations [41, 42]. It is for these reasons we aimed to assemble conversations of people living with MS and their families about ACP which are occasionally discussed but rarely voiced and to juxtapose them with health professionals' views to increase the audibility of their narrative. Our specific research questions were to understand under what situations, with whom, how, and why do people with MS (and their families) engage in ACP and what works for whom, how, and why, during ACP discussions?

## Method

### Study design

Our study made use of multiple data sources using different qualitative methods to understand the complex and nuanced issues associated with ACP relevant to people living with MS and their families from multiple perspectives. First, we conducted interviews with people living with MS and their family members to sensitively tease out the concepts, ideas, frameworks and structures of meanings associated with ACP. This approach was appropriate for our study because it involved qualitative methods of data collection that were minimally intrusive allowing us to gather data compassionately [43] given the potential vulnerability of participants [44]. We then conducted what we refer to as an 'ethical discussion group' where health professionals were invited to participate in a group discussion to examine their reasoning and justification of ethical principles underlying their negotiation of planning, including ACP, with people living with severe MS. Previously, data generated using this approach have helped identify complex ethical, legal, and clinical issues that practitioners experience in assisted reproductive services, embryology, stem-cell research, and solid organ donation [45–48].

## Setting and participants

The study was conducted across two main MS services, the first located in southeast London with an estimated 3,000 people with MS, and the second serving west London, Surrey and East Berkshire with an estimated 3,500 people with MS. Eligibility criteria for inclusion in the study included those who were adults aged 18+; having a confirmed diagnosis of MS; an Expanded Disability Status Scale (EDSS) [49] score of $\geq 6$ suggestive of severe disease (requiring walking aid(s) to walk, to those restricted to a wheelchair, or bed at the upper end of the scale) and; being able to provide informed consent.

## Sampling and recruitment of people with MS and their families

The interview component of the study intended to purposefully recruit 30 people with MS up to a point where data saturation would typically be realised. Potential participants were prospectively identified by consultant neurologists at three hospital sites during outpatient clinics to reflect a range of clinical and demographic characteristics relevant to the study, including age, ethnicity, gender, MS-type (relapsing-remitting, primary progressive and secondary progressive) and EDSS score. Potential participants were provided with a study information sheet, developed in collaboration with our patient and participant involvement (PPI) representatives. Written informed consent was obtained before each interview. In some instances, family members needed to be present during the interviews and we refer to these situations as dyads. Where family members were not present during the interviews, participants were asked permission for us to contact them to obtain their perspectives on ACP up to a total of 15 family members. Family members identified through study participants were provided with a tailored information sheet and informed consent was obtained before interviews commenced.

## Sampling and recruitment of professionals for the ethical discussion group

Potential participants for the ethical discussion group were purposefully selected to represent a range of specialties (neurology, palliative care, rehabilitation medicine and primary care) and professions (doctors, clinical nurse specialists and physiotherapists) working with people affected by MS and their families.

## Ethical approval

Ethical approval was provided by Camden & King's Cross Research Ethics Committee IRAS project ID 258274 REC reference ID 19/LO/0292.

## Data collection—interviews

Data collection took place in participants' homes. The interviews were conducted by LC and CP, two female health services researchers with considerable experience in palliative and end of life care research. They used a topic guide (refer to S1 Appendix) developed in collaboration with the study's PPI representatives and were audio-recorded. Interviews commenced by breaking the ice [50] to develop rapport with participants. In the first instance, participants were invited to tell their story of their illness in three phases: the past, present, and future. This led to exploration about decision-making, planning for future care and their understanding of ACP. Flexibility in the interview guide allowed the interviewers (LC and CP) to carefully navigate difficult topics raised within the interview and take rest breaks as appropriate. Interviews lasted between 30 and 126 minutes (median 73 minutes).

## Data collection—ethical discussion group

Before the ethical discussion group took place participants critically reflected upon our realist review [32] aimed at developing and refining an initial theory on engagement in ACP for people with MS. Participants also considered key findings from the interviews with people with MS and family members. The ethical discussion group was facilitated by BF, a bioethicist. Due to circumstances associated with the COVID-19 pandemic, the group was hosted online (via Zoom) for 1.5 hours and was audio-recorded. Data collection aimed to elicit attitudes views and ethical reasoning on the content, place, timing, and challenges involved in discussing future planning and specifically ACP among people with MS and their families, to explore participants' reasoning and justification for their beliefs, practices and ethical principles underlying their negotiation of ACP, to identify challenges when engaging people with MS and their families in this process and to consider solutions to these issues (refer to S2 Appendix for the topic guide).

## Data management and interpretation

The data analysis process was undertaken in two distinct stages, the first stage involved data management using the framework approach [51] to facilitate the second stage, interpretation. Data management was led by CP and JK and began part way through the interviews. Data management involved data familiarisation to develop a thematic framework developed inductively from the interviews and deductively based on specifically, context-mechanism-outcome configurations from our realist systematic review (present in Table 1) [32]. The resulting framework was informed by multiple discussions between CP and JK and was tested and revised following independent coding of four transcripts where early emergent findings (based on phase 1 interviews) were discussed and validated with members from the project advisory group and PPI group. CP 'charted' data from each interview across themes and corresponding sub-themes into the framework matrix. Charting was facilitated using NVivo 12 software. The matrix facilitated both case and theme-based analysis, exploring within cases and groups of cases, and within and across themes.

We adopted a realist approach to analytical rigour [52, 53] using criteria that researchers typically agree constitutes high quality qualitative analysis [54, 55]. The criteria we adopted and our actions are presented in Table 2.

## Results

### Sample characteristics

In total eight people with MS who were referred by the clinicians to the research team declined to take part in an interview. The reasons for this included feeling too tired and not wanting to talk

**Table 1. 'Context-mechanism-outcome' hypotheses [32].**

| | Context | Mechanism | Outcome |
|---|---|---|---|
| 1. | If people with MS experience losses | then they will accept that MS is life-limiting and will come to see themselves as a person with a life-limiting illness | and they will be more likely to engage in ACP |
| 2. | If people with MS have a trusting and empathic relationship with their healthcare provider | then they will feel empowered | |
| 3. | If people with MS feel they are a burden to family members | then they will look for ways to reduce their family member's future decisional conflict, | |
| 4. | If people with MS want to establish control over their future, | then they will come to understand ACP as a tool for autonomy | |
| 5. | If health care professionals have the communication skills to engage in open, frank, and timely discussions | then this would inspire the confidence to discuss death and dying | which would facilitate ACP engagement and completion. |
| 6. | If people with MS have witnessed 'bad deaths', then they will fear dying | and will perceive ACP as a way to prevent a 'bad death' | thus, will be more likely to engage in ACP. |

**Table 2. Quality criteria selected for ensuring rigorous qualitative analysis [53, 54].**

| Quality criteria | How it was achieved |
|---|---|
| Rich rigour—analysis uses appropriate sample, context and data-driven by theory | We collected data from 27 people living with MS and 17 relatives and five health professionals working with people living with MS and other life-limiting conditions. Interviews were semi-structured and provided scope for participants to tell their stories in their own words. We drew on the 'context, mechanism outcomes' derived from our realist review [31] to facilitate analysis of the primary data |
| Credibility and authenticity–thick descriptions and detailed findings have been provided to support inferences | We believe a wealth of qualitative data derived from multiple qualitative data provide for 'thick description' and detail that describe the highly complex and nuanced situations surrounding ACP for people living with MS and their families. We reflect on the experiences of the participants as they lived them and perceived them. Quotes were selected from a range of participant interview transcripts. |
| Criticality—detailed account of how researchers critically appraised their findings | Each stage of the analytic process is outlined clearly. During analysis, the two researchers (CP and JK) responsible for data analysis offered critical and alternative interpretations and explanations of findings, regularly challenged each other's assumptions, and encouraged frank and open introspective discussions (for example how each researcher's biases, experiences, and histories impacted the analytic process, particularly against the backdrop of the COVID-19 pandemic). |
| Attention to contradictory or non-confirmatory data | During analysis, CP and JK were mindful to pay attention to data that contradicted or questioned the narratives of the main emerging themes and incorporated them into the subsequent development and in the reporting of data. |
| Fidelity or meaningful coherence—analysis achieves its intended goals through using appropriate methods | To realise our study question we developed a 'thread' that would hold the study together commencing with our recruitment strategy, topic guide, interview style, analysis plan, reporting of findings and their interpretation of the findings alongside the CMOs tested in our realist review and wider literature. |

about the future among other issues. Table 3 provides details of 27 people with MS and 17 family members who participated in the study. Interviews were conducted between June 2019 and March 2020, just before the first UK COVID-19 pandemic lockdown. Depending on participants' preferences, 21 interviews with people with MS took place independently and in 10 instances where the person with MS and their family members were interviewed together in dyads.

People with MS included 16 females and 11 males aged between 38 and 75 years (median 59 years). A minority of participants (n = 4) were from a Black, Asian or ethnically diverse background. The majority were living with a spouse or partner while five lived alone and three with parents. The sample included people with both primary progressive and secondary progressive MS as well as four participants with relapsing-remitting MS. A range of EDSS scores between 6 and 8.5 was represented. Family members taking part included 11 females and six males, aged between 31 and 77 years (median 65 years). Most were the spouse or partner of their dependant. Three parents, two siblings and one adult child also participated.

## Emerging themes from interviews with people with MS and their families

Three main themes emerged from the analysis of interviews with people with MS and their family members and included (i) planning for an uncertain future; (ii) perceived obstacles to engage in planning and (iii) preferences for engagement in ACP.

**Table 3. Characteristics of participants.**

| Characteristics of people with MS | n |
|---|---|
| **Sex:** | |
| female | 16 |
| male | 11 |
| **Age:** | |
| median years (range) | 59 (38–75) |
| **Self-assigned ethnicity:** | |
| White British | 23 |
| Asian/Asian British | 1 |
| Black/African/Caribbean/Black British | 2 |
| Other ethnic groups | 1 |
| **Living arrangements:** | |
| alone (supported/LTCF) | 5 |
| with spouse/partner | 19 |
| with parents | 3 |
| **MS type:** | |
| relapsing-remitting | 4 |
| primary progressive | 13 |
| secondary progressive | 10 |
| **Years since diagnosis:** | |
| median years (range) | 18 (3–39) |
| **EDSS score:** | |
| 6–6.5[a] | 5 |
| 7[b] | 6 |
| 7.5[c] | 7 |
| 8–8.5[d] | 9 |
| **Characteristics of caregiver participants** | **n** |
| **Gender** | |
| male | 11 |
| female | 6 |
| **Age** | |
| Median year (range) | 65 (31–77) |
| **Relationship to person with MS** | |
| spouse/partner | 11 |
| parent | 3 |
| sibling | 2 |
| adult child | 1 |

EDSS score interpretation:

[a] Requires walking aid(s) to walk;

[b] Essentially restricted to a wheelchair, can transfer alone;

[c] Restricted to a wheelchair, may need aid in transferring, may require a motorised wheelchair;

[d] Essentially restricted to bed or chair.

**1. Planning for an uncertain future.** Whilst none had made, or in many instances were not familiar with ACP, more than one-third of participants had engaged in one or more activities associated with planning their care and treatment. For example, they had engaged in more formal planning tools including Lasting Power of Attorney for health and welfare (LPA) (n = 9); advance decision to refuse treatment (ADRT) (n = 2); and do not attempt cardiopulmonary

resuscitation (DNACPR) orders (n = 4). The motivations for engaging in each were different. In our study, LPAs were made by people with MS with a range of EDSS scores. Family members of people with MS (including spouse/partner, siblings, parents, and adult children) were often appointed as the 'proxy decision-makers entrusted with making health and care decisions on their behalf. In all cases where people with MS had made an LPA for health and care decisions, they had given their proxy decision-makers authority rather than their doctors or other members of the clinical teams, to refuse or consent to treatment, including life-sustaining treatments. For people with MS with the highest EDSS level (8–8.5), a motivating factor for families to support their dependents making an LPA was guided by their progressive loss of cognitive capacity.

> *"Making an LPA was triggered off because I wanted to be able to make decisions on his health because he can't say for himself what he wants. And when his memory was getting worse, I thought I'm gonna get that done."* (K003, spouse/partner of male PwMS, 56–65 age group, primary progressive, EDSS 8–8.5)

Family members of dependants who had fluctuating cognitive capacity described the circumstances concerning how they arrived at a point where planning became necessary. In one instance this was prompted by the content of a TV programme leading to a discussion to understand their dependant's 'in-the-moment' wishes. Inevitably, this necessitated being flexible to respond to the uncertain trajectory of their MS. For example, decisions, whether to refuse or consent to life-sustaining treatment, fluctuated over time, typified by the following.

> *Relative:* "*I said to '[person with MS] can I just ask you a question?" "If something happened to you to the point that you were very unwell, what would you choose?"*
>
> *Interviewer: And what did you say [person with MS]?"*
>
> *Person with MS:* "*I would want to live."*
>
> (K003, spouse/partner and male PwMS, 56–65 age group, primary progressive, EDSS 8–8.5.)

However, another relative of a dependant with secondary progressive MS shared a different and contrasting narrative.

> *"I think the way she is now I would say she'd want us to help her go. So, it all depends. I think she wouldn't just want to be sitting around and to be a vegetable because she couldn't cope with that."*
>
> (K004, spouse/partner and female PwMS, 56–65 age group, secondary progressive, EDSS 8–8.5)

In other cases, participants were motivated by the experience of an acute event following complications associated with their MS, for example, urosepsis or aspiration pneumonia, where they became too unwell to communicate their wishes. This prompted them to ensure they identified a proxy who would faithfully advocate for them typified by the following.

> *"I wanted to give the people with me, primarily [partner] and friends, mainly [partner], permission to say, "No, I have power of attorney, so this is what we're gonna do . . . that's what [he] want[s]." "I don't want there to be any ambiguity."*
>
> (K001, male PwMS, 56–65 age group, secondary progressive, EDSS 7)

For many, the legal basis of the LPA in the UK was a motivating factor. Participants felt it was important to formalise the right of the family to be involved in decisions regarding their health and care so that they were considered for example.

*"[If I] get in chronic difficulties, I want everyone to know my wishes. . . and I've talked about them before, but they're not formalised. It's not legal. . . I was writing my will, and the solicitor mentioned LPAs. . . I thought well, it's probably a good idea to make one."*

(K011, male PwMS, 36–45 age group, secondary progressive, EDSS 8–8.5)

Although in some instances professionals suggested patients make an LPA for property and finances, interestingly LPAs for health and welfare were never discussed. Therefore, when this type of LPA had been drafted and registered independently by participants some expressed concerns health professionals would not be aware that they possessed one and worried if their wishes would be honoured.

*"I've done an LPA. Now the thing it doesn't do is communicate that to the professionals. An advance care plan probably would because they would have access to that. They can't see in my filing cabinet, so they won't know my wishes. I guess having it formalised and on file some-where with the lot would definitely be beneficial, otherwise, they're guessing."*

(K0004, male PwMS, 36–45 age group, secondary progressive, EDSS 8–8.5)

Typically, participants made LPAs without employing a solicitor. However, they described it as a "challenging" "tedious" and expensive process, involving a not insignificant cost, particularly if subsequent iterations were required.

*"It was really hard making our LPA, but it got progressively easier with each form we did. We sort of got a better handle on what it was they wanted and how they wanted you to do it and things. But you can't help feeling it could probably be a bit easier still. And they've reduced the cost of them, which is another good thing because I think we paid £160 for each of them. A lot of money."*

(K007, female PwMS, 56–65 age group, secondary progressive, EDSS 6–6.5)

There are also challenges in making an LPA for those without family members. For exam-ple, a participant who lived alone wanted to ask a friend to be her proxy decision-maker but was unsure how to commence the conversation. Despite this, participants felt it imperative to have an advocate to rely on rather than professionals to make critical decisions on their behalf.

*"I haven't done it yet, but I . . . I don't know why I haven't done it yet, but I want to make [my friend], my power of attorney . . . Because I know now, that while she might not like my deci-sions, she will do what I want . . . And that's so incredibly important. . . if I'm not being lis-tened to then I've got an extra voice."*

(LG003, female PwMS, 56–65 age group, primary progressive, EDSS 7.5)

*Advance decision to refuse treatment (ADRT).* Two participants, both in the highest EDSS level group of 8–8.5, made an ADRT, not wanting further invasive and potentially futile life-sustaining treatments. Both believed that while they were able to enjoy aspects of their life, they were adamant this was outweighed by the considerable challenges resulting from their

MS. These included chronic pressure ulcers, double incontinence, speech difficulties and percutaneous endoscopic gastrostomy (PEG) issues. One described feeling 'useless' and believed she was a burden to her family,

> "I don't think it's worth making the effort with me really. I think I may be finished. You've got to finish sometime, and it may be that I've reached a natural conclusion. I've brought the children up; I've done most things that I want to do... I don't want any more fiddling about like this, I often feel, 'Just leave me alone', it's part of that... For me and [spouse/partner] and the kids, especially for me, [the ADRT is useful... I don't want to hang on uselessly, make life so difficult for everybody else."

(LG015, female PwMS, 66–75 age group, primary progressive, EDSS 8–8.5)

However, a family member pointed out that trying to apply an ADRT in a crisis was not straightforward, especially when his dependant experienced periods of relative stability followed by an "abrupt" episode of acute illness. He worried about the various scenarios where an ADRT might apply. In this case, while his partner had made an "exceptionally good" recovery from pneumonia, another quite traumatic experience in ICU galvanised a decision to avoid any future admission to the hospital.

> "The doctor, because of that ADRT, um, said, "Should we apply oxygen?", so I then had to think... it was difficult for me to try and interpret that... once you're actually faced with the practicality then it's not quite as clear cut as when we first got it drawn up. It's not necessarily black and white when you write it down. You can't think about all circumstances are going to be. I just don't think you can sensibly paint all the scenarios without making the person just go bonkers."

(LG015, spouse/partner of female PwMS, 66–75 age group, primary progressive, EDSS 8–8.5)

*Do not attempt cardiopulmonary resuscitation (DNACPR) order.* Both participants with ADRTs had additionally made a DNACPR order. This appeared to be motivated by a wish to have an all-encompassing solution to ensure that their decisions to refuse treatment were immediately visible to clinicians.

> "Having spoken to people in the hospital we thought that a DNACPR was an extra, you know, if you get an ambulance called or something, they might be more noticeable."

(LG015, spouse/partner of female PwMS, 66–75 age group, primary progressive, EDSS 8–8.5)

In the two other cases where a DNACPR order had been made participants changed their minds and withdrew, or planned to withdraw, their decision. These two cases highlight the importance of exploring the motivations of people with MS when making DNACPR orders; the desire to make this decision may be indicative of other underlying issues that would benefit from appropriate support for example depression or unmet care, or equipment needs.

> "When things were going wrong, I made sure that it had on my form 'Do Not Resuscitate'... As soon as these people [carers] came along, then I took it back... And I've decided that I wanted to live... That I could have someone to talk to her [care home] that cared... I decided well, I'll live. I'm not sure I agree now I'm not sure I would keep it on if anything more

*happened to me. . . I still suppose the DNACPR is in place. . . but I don't want them to purposely let me go."*

(LG005, female PwMS, 66–75 age group, primary progressive, EDSS 7.5)

*"I didn't tell my husband and then I thought well I better tell him, and he said, "Don't do that to me!". I went back and spoke to my GP, and she said, "Why do you want to change your mind?" and I said, "Because of my husband, I couldn't do it to do him, it was horrible . . .how could I do that to him? I mean if it happens, it happens, but I changed my mind about the DNACPR. I've been thinking a lot about it, and I changed my mind."*

(LG012, female PwMS, 56–65 age group, secondary progressive, EDSS 8–8.5)

**2. Obstacles for engagement in ACP.** While more than one-third of participants understood the principles of ACP and engaged in some type of formal planning-related activity, most had not. The interviews brought into sharp relief several barriers to engagement in this activity that included uncertainty relating to MS, negative experiences of ACP-related discussions with health professionals, familial relationships, transitioning from disease-modifying treatment to supportive care and prioritising symptom management and quality of life over intermediate and long-term planning. A number of these issues are enriched by and have correspondence with the views shared by participants involved in the ethical discussion group.

*Uncertainty relating to the MS illness trajectory.* Many participants with MS and their family members' attitudes towards ACP and its relevance to them, were influenced by their experience of the MS illness trajectory as being inherently uncertain. Attitudes towards ACP in the context of an uncertain illness fell into three main groups. For one group, fear of an uncertain future acted as a barrier to engaging with ACP. Participants balanced their awareness of possible illness trajectories of increasing disability with the hope it would not apply to them. Some readily admitted their coping strategy to live alongside an overwhelming situation was to stick their 'head in the sand' or deceive themselves about their condition for example.

*"I don't think I can plan for my future because I don't know what's going to happen. Just because I've got MS it doesn't mean to say I've got the same MS as the lady around the corner. She might have experienced different things. I just feel, um, I just wanna live and not think about the future. I'm scared because I don't know what it's going to be like."*

(LG009, female PwMS, 46–55 age group, primary progressive, EDSS 6)

For another group of participants, while accepting their MS was progressive, this reality was counter-balanced by a belief they would continue to successfully adapt to the multiple losses associated with their condition.

*"For MS, it's different because most of us don't feel like we're dying. We are losing the ability to do things. . . I have a friend with MS. . . she probably hasn't been able to stand up and walk for two years. . . she's had sepsis as well, but she's starting to do standing classes, so you know she's not giving up."*

(K005, female PwMS, 46–55 age group, relapsing-remitting, EDSS 7)

Additionally, some contrasted their illness with other life-limiting conditions such as metastatic cancer or MND where death was an inevitable result of their advanced disease. MS, however, was different as evidenced by the following.

*"You think about cancer, where you're dying in the next few weeks. But we're in a bit of limbo-land. He's had three stays in ICU, where we have been brought into a room and it has felt like, if he doesn't get intubated, he might die in the next two hours. That's how it's felt. And then yet three weeks later he's at home going to the park, having ice cream. And smiling and holding my hand and laughing."*

(GST001C2, sibling of male PwMS, 36–45 age group, primary progressive, EDSS 8–8.5)

The third group of participants felt ACP might support their wish to be optimistic and were more positive about planning. For these participants, engaging in the interview prompted them to consider the relevance of ACP, reduce their anxiety and ensure their families were not unduly burdened if they became unable to communicate them, illustrated by the following.

*"They're grown-up things, they're scary grown-up things, and they should be addressed . . . And I think it would be helpful, it's one of those things that kind of lurks at the back of your mind. I haven't quite addressed them. . . I suppose in a way it's not for yourself; it's for people looking after you or that love you or whatever, so they don't have to have the burden of making the decision."*

(K008, female PwMS, 56–65 age group, secondary progressive, EDSS 7.5)

*Poor experiences of ACP discussions with clinicians.* A second barrier to engaging in ACP was associated with previous unhelpful experiences of discussing the future with clinicians and in some instances with those working in palliative care. Participants described encounters they felt inadequately prepared for and where the content of conversations did not align with their values or preferences. Consequently, some questioned the intentions and motives of clinicians who they believed made incorrect assumptions about their quality of life, level of disability, age and what was important to them. They were left feeling threatened with an overarching suspicion their care was going to be rationed or curtailed, typified by the following.

*"There's one thing that you've really got to be careful about. I was in hospital and had yet another doctor who wanted me to sign, wanted me to agree to a power to [refuse treatment] . . . it's kind of really dodgy, you know, you know [laughs], I'm very suspicious now."*

(GST002, male PwMS, 46–55 age group, primary progressive, EDSS 8–8.5)

*"I argued with one of the doctors in hospital because they had it on my record, Do Not Resuscitate and I said, "I don't remember ever saying that!", he said, "Oh, we'll change it back then", so I said' "Yeah you will!", 'But I never said I didn't want to be resuscitated'*

(LG014, male PwMS, 56–65 age group, secondary progressive, EDSS 7)

For some participants with an LPA in place, negative experiences of ACP-related discussions with clinicians led to an erosion of trust and disengagement with services. For example, where participants had agreed to a palliative care referral for additional supportive symptom care, they quickly became wary of early uninvited discussions about end-of-life planning including decisions to refuse life-sustaining treatment. Their efforts were focused on experiencing a better quality of life, not preparing to die. Consequently, some participants, illustrated by the following comment, disengaged with palliative care, thus losing potentially valuable support.

*"I said to the palliative care nurse, "Please stop asking him about resuscitation! You've not mentioned anything about palliative care and yet you come straight up to him with things like that." So, she just sort of said, "Oh well, if he gets any worse you know where to find us." and I thought, I won't find you."'*

(K003, wife of male PwMS, 56–65 age group, primary progressive, EDSS 8–8.5)

*Family tension.* Just as we observed obstacles to discussing ACP between professionals and people with MS, they existed within families too. For example, participants believed discussing the future might upset their family members. There was sometimes a reluctance to engage in these challenging conversations that some believed were just too painful.

*"I'll say that if I sat her down and discussed ACP it would have a negative effect. She would think that I'm trying to get rid of her. She would see herself, you know, looking at death's door. We do not talk about subjects like that, in my view for fear of getting her maudlin and depressed."*

(LG001, spouse of female PwMS, 66–75 age group, secondary progressive MS, EDSS 7.5)

However, family members also acknowledged the necessity of having these conversations and not making assumptions about their wishes should they become unable to communicate or decide themselves.

*"I think it's so easy to assume that you know. To imagine that I know how he feels about everything. But I don't think I should. I can never really assume that I know those things. I think it's good for me to be clear... like what would he want, but then if, in a different situation he might give different response. So, it's good for me to be clear on how he sees everything."*

(GST002, sibling of male PwMS, 46–55 age group, primary progressive, EDSS 8–8.5)

*Lack of information about additional or availability a supportive care pathway.* Planning was stymied by participants not knowing they had been transitioned from disease-modifying treatment to supportive care. Since there were frequently no explicit discussions about their disease trajectory, participants described 'guessing' they had arrived at a new destination. This information void about the future, intentional or otherwise, and in some situations contrived by parties to avoid making reality explicit, contributed to their reluctance to think about their future or consider in what ways ACP might be relevant or helpful.

*"When you get diagnosed as [secondary] progressive MS no professional mentions it's life-limiting in terms of your longevity."*

(K011, male PwMS, 36–45 age group, secondary progressive, EDSS 8–8.5)

*"I have wondered about the swallowing process. What if I can't swallow? I haven't had any information at all... I don't physically know what they would do if I couldn't ever swallow again... I haven't even mentioned it to my neurologist, to be honest. I'm not sure what happens actually. I think you have some sort of feeding tube or something like that, don't you? ... I ain't got a clue."*

(LG009, female PwMS, 46–55 age group, primary progressive, EDSS 6–6.5)

*Prioritising symptom management, rehabilitation needs and quality of life issues.* It is important to note many participants' priorities were focused on optimising current living and quality of life rather than thinking about the distant future. These issues often took precedence during hospital appointments meaning there was little opportunity to discuss the future including ACP with professionals. Moreover, participants also faced long waits for assessment and provision of services to support their activities of daily living. Negotiating access to services was time-consuming and exhausting meaning ACP became a low priority typified by the following.

*"Once a year you see the neurologist, so you tend to have your little mental list. You're never going to get to advance care planning, which is a shame."*

(K007, female PwMS, 56–65 age group, secondary progressive, EDSS 6–6.5)

**3. Preferences for engagement in ACP.** Participants (including those who had made LPAs), nevertheless, explained they were willing to discuss ACP with their MS care team and stated MS nurses were the most appropriate professionals to introduce this topic. There was also broad agreement that initiating ACP-related discussions should be undertaken by those trained in empathic conversations and who had a trusting relationship with them and their family. The timing of discussions needed to be bespoke although there was consensus that sharing news of a transition to secondary progressive MS was an appropriate time to introduce ACP.

*"You've got to have some very gentle way of approaching it, whereby people don't have to face the stark reality if they don't want to. There's got to be a middle way somewhere . . . just a gentle way of starting the conversation should someone want it, that's what needs to be found in my opinion. Round about that, you know, change of relapsing-remitting to progressive, I think that would be a good time to broach it."*

(K011, male PwMS, 36–45 age group, secondary progressive, EDSS 8–8.5)

*"I know a lot of people with MS get depressed. It (ACP) would have to be done very tactfully indeed and at the right time. But if it can be introduced as a sort of vague thing people can kind of mull over it a bit and come back to it when they want to."*

(K008, female PwMS, 56–65 age group, secondary progressive, EDSS 7.5)

## The ethical discussion group

Five health professionals consented to participate in the ethical discussion group including an MS clinical nurse specialist, consultant neurologist, consultant in palliative care, consultant physiotherapist and a general practitioner. The ethical discussion group revealed a shift of thinking in professionals' previous attitudes and beliefs regarding ACP, all of which had implications for their current practice and how they wished to engage in this process. Specifically, those working in palliative care and general practice shared that earlier in their careers they saw ACP as an appropriate and useful intervention where a formal, structured approach was essential; they later became doubtful that in this format it conferred benefits that accorded with patients' preferences (refer to Table 4 for emerging themes and illustrative examples of health professionals' narratives). They noted that since MS is a long-term neurodegenerative condition, if ACP was pursued, it should be refashioned to become a process of ongoing 'meaningful dialogue' and review built on a trusting, empathic relationship with patients and

**Table 4. Emerging themes and illustrative quotes from the ethical discussion group.**

| Theme | Illustrative quotes |
|---|---|
| **A shift in practice** | *"When someone mentions ACP, I have quite a different reaction now than I did early on in my palliative care career. So perhaps at the start of my career, I would have thought that it's an excellent thing, we all need to be focusing families and patients on actually doing one, then having it properly written down and cascading that to all the relevant professionals. But right now, I feel a little bit in the middle about it. . . I'm less militant about it, I think now."* (Consultant in Palliative Care) |
| | *"Early on you can get caught up in this very sort of 'tick box' approach to ACP that we must do it and it's a good thing. Actually, what I've come to sort of reflect on as I've gone through my time, is it's the quality of the conversations that matter really and um, often even writing it down is a pointless exercise at times and we're often very paternalistic and we medicalise ACP* (General Practitioner) |
| **Relevance of ACP for people with MS** | *"The idea of pushing ACP is quite difficult. . . We use it [ACP] a lot in people who are very badly affected by strokes, but in MS I think it's a bit of a different deal."* (Consultant Neurologist) |
| | *"I feel the approach for long term. . . conditions, needs to be different from those with a very clear trajectory. . . with long term conditions there needs to be a framework of walking alongside a patient . . .."* (Consultant in Palliative Care) |
| **Less medicalised and formalised approach to ACP for people with MS** | *"Things like lasting power of attorney, I think there's sometimes misunderstandings of what that means, by patients. . . And I think there's also a misunderstanding of what that means by professionals as well. . . so from both perspectives, there's a lack of clarity and perhaps we need a bit more discussion around that and what it means"* (Consultant Physiotherapist) |
| | *"We need to see what type of people are, they are, the patients or the families, and how they deal with life in general. Are they risk-takers, are they not, are they planners, are they not? Also, the person being able to just state in a value statement, that, 'I do value life, my life is worth living' if they felt that they have said that then they can trust the doctor to make the right decision for them, 'they've heard my voice and my voice said, my life is worth living, even though I am PEG-fed."* [percutaneous endoscopic gastrostomy] (Consultant in Palliative Care) |
| | *"A statement of wishes, as a clinician, would be far more useful to me. Tell me who you are as a person, tell me what's important to you. . . If I was having to decide, you know, for somebody because they couldn't decide, that's what I would want to know about them."* (General Practitioner) |
| **ACP challenges for people with MS** | *"There was a patient I remember particularly because I had her, her wishes, written down and nobody knew about them, so you know it's like what's the point in having them written down if, you know. . . it's really difficult to get control at that moment when. . . you've got people running around doing stuff to people or not doing stuff to people, in the middle of the night, and again, frightening, not just because people aren't listening to them"* (Consultant Neurologist) |
| | *"There's a huge degree of uncertainty about the trajectory that that illness is going to follow and we don't like uncertainty, and it's very difficult to broach that uncertainty and to admit to that uncertainty with a patient, and it doesn't feel, you know when you diagnose somebody with [a long-term condition], I don't know anybody who would. . . start having ACP conversations with somebody at that point* (General Practitioner) |
| | *"I have to say that when ACP comes up. . . I do have a sense of panic in me thinking, oh, my goodness, I hope it's not me who will have to be involved with that, and I do completely feel unprepared for having a conversation like this with patients* (Clinical Nurse Specialist) |

their families, rather than being employed as a 'one-time' event or 'tick-box' exercise. The consultant neurologist believed this was best facilitated in the 'spaces' around the usual care associated with practical, clinical, and day-to-day physical and psychological needs and this relied on getting to know the person on their terms. What is important to note is that this broader, less formal understanding of ACP may not have been adequately communicated to patients. All ethical discussion group participants acknowledged that patients' priorities were likely to be fluid, largely due to the clinical uncertainty associated.

Interestingly, and at odds with some of the narratives of participants with MS, some ethical discussion group participants expressed concerns about the legally binding aspects associated with ACP. They stated that thinking through all the possible future scenarios where an ADRT might apply to people with MS were inherently challenging. They also shared their hesitancy in knowing how LPAs for health and welfare were operationalised. This might account for why participants with MS stated they were rarely discussed. Instead, ethical discussion group participants emphasized that developing an understanding of a person with MS's values and preferences as being fundamentally more useful in supporting shared decision-making that did not automatically include discussions about preferences for life-sustaining treatments.

The open and frank nature of the group provided agency for some participants to be self-critical and voice their deficiencies and apprehension in knowing when and how to commence ACP conversations. Some spoke of the time demands associated with discussing this topic when the physical needs of their patients were sometimes very complicated. Compressing delicate conversations into the all too brief appointments was acknowledged and this troubled them. Whilst the clinical nurse specialist, suggested by the people with MS as being the most appropriate professional to have these conversations with, exposed his/her lack of confidence in the skills necessary to initiate the carefully worded dialogue with patients, a fear of getting it wrong and causing harm. Another key challenge was associated with how ACP conversations were shared with other professionals. For example, a consultant neurologist acknowledged that including details obtained less formally during ongoing dialogue with a patient about their values and preferences in a clinic letter may not be the most appropriate way to communicate valuable information, accessible to all relevant parties. The GP participant went further to acknowledge the difficulty in capturing a patient's values and preferences expressed during the 'snippets' of conversations accumulated over many months and more often years in a single summary care record. Participants also described situations where written statements or legal documents relating to ACP were not accessible when needed during a medical crisis which helped no one.

## Discussion

This study provides a detailed qualitative understanding of the place and meanings of ACP for those living with MS and their families and contrasts these accounts with those of the health professionals who care for them.

### Strengths and limitations

A strength of this study is the detail in which we were able to explore the narratives of people living with severe MS about a sensitive topic, exploring notions of loss, compromised mental capacity and intimations of mortality. Our participants could therefore be described as inhabiting positions of 'intrinsic and situational' vulnerability that puts some at greater risk of being exploited, a justification to exclude them from the research process [44]. However, not engaging with them compromises the ethical principle of justice, particularly when the research at hand spoke so directly to participants' needs and experiences. This study also made use of a

multi-data qualitative approach. Specifically, the simultaneous design enabled findings from our realist review [32] and interim interview findings from the interviews to be critically considered by ethical discussion group participants that produced a more detailed understanding of the multiple realities associated with ACP.

We acknowledge this study has limitations. First, the interviews were cross-sectional. Whilst study participants were actively encouraged to reflect retrospectively on instances where their views and decisions were triggered by events, we recognise people with progressive diseases for example MS, have, views and needs that change over time. Repeat interviews enable researchers to access and make sense of competing and often contradictory accounts and to understand why views change and have successfully been employed in palliative care research, mostly among those with cancer [56–58]. Second, data on the experiences of 10 people with MS were obtained with their family members being present rather than talking to them alone. Ideally, people with MS would be consulted directly, but this is not always possible, often because their disabilities, e.g., communication difficulties, prevented it. However, in our study and for those with people with communication impairments, family members' contributions represented a valuable additional source of data to aid our understanding of the experience of people with MS. Third, we are aware we recruited few participants from Black, Asian and ethnically diverse communities. This is a cause of concern for a study set in within a metropolitan area where ethnic diversity was present. Evidence suggests that attitudes to truth-telling, filial responsibility, notions of patient autonomy and requests for 'aggressive' treatments may be culturally patterned and influence shared decision-making, including ACP [59–61]. Linked to this concern, we did not explore study participants' perspectives about religiosity and spirituality. Previous research has demonstrated that religiosity has an influence on the relevance and acceptability of advance care planning [62, 63] which may also be present among those living with MS [64]. Future research should incorporate a focus on these issues. Last, this study reached theoretical saturation just before the first COVID-19 pandemic lockdown in the UK. The pandemic placed many vulnerable patient populations including those with MS, at risk of responding poorly to COVID-19 [65] where there was a greater emphasis on engaging in ACP for vulnerable populations, including those with MS [28–31, 66]. Although the ethical discussion group participants mentioned practical challenges associated with engaging in ACP-related conversations virtually during the pandemic, we omitted to include perspectives of people living with MS and their families during this critical moment in time. Some may have experienced anxiety from the pandemic [67] amplified by conversations regarding ceilings of treatment and this would be useful to understand.

## Main findings

Rather than not wishing to consider their future, we present accounts from many participants who wished exactly to do this, often in concert with their families, albeit with caveats. Their accounts are nuanced and sometimes contradictory which reflect the complexities of ACP [68], a situation amplified with neuro-generative conditions like MS [35]. Below we appraise our study findings alongside the 'context-mechanism-output' configurations [69] that were tested during our realist review of ACP for people living with MS and their families [32] and wider literature.

### Recognition of punctuated losses

For some people with MS, punctuated losses associated with clinically significant disease events [40, 70–73] that may be associated with positive coping strategies [5] leading to acceptance of MS as a progressive condition and the creation of a new self-identity where ACP may

become relevant. While participants in this study recognised health crises resulting from complications associated with MS could potentially result in death, they did not necessarily equate their situation with dying. Consequently, some were reluctant to consider ACP planning strategies including DNACPR or ADRTs as being relevant to them. This observation is at odds with findings from those with MND, another progressive neuro-generative disease, where ACP was considerably easier for those who accepted encroaching death [74] although it was also present at other stages of the disease, albeit to a lesser extent. Instead, we observed participants were willing to engage in drafting and registering health and welfare LPAs to ensure that a family member could advocate for them if they were ever too sick to communicate themselves or lost cognitive capacity. Consequently, life-sustaining treatments were viewed as appropriate in the context of a treatable acute health issue such as sepsis following a urine infection or aspiration pneumonia. Moreover, having someone to advocate on their behalf brought considerable comfort to participants and family members. The flexibility of an LPA was felt to be important by participants in the context of a fluctuating and uncertain illness and yet was never suggested to them by health professionals.

There were two cases where participants had made an ADRT, principally because they felt their quality of life was too low to warrant aggressive, potentially futile treatment and they believed they were a burden to family members. However, they acknowledged they could conceivably live for many more years. For one, feelings of being useless were present. Identifying which people living with MS are less able to develop strategies to cope with their situation especially when uncertainty is omnipresent [5], should lead health professionals to address their potentially modifiable state with psychological interventions, for example, cognitive behaviour therapy [75]. However it must be noted some individuals may perceive this as a rational assessment of their current state and potential future [76].

## Value of trusting, empathic relationships with known skilled health professionals

Building a trusting, empathic relationship with a known health professional centred on active and reflective listening and validating patients' concerns and fears is foundational for engagement in ACP [8, 37, 77–80]. This sentiment was evident among health professionals who participated in the ethical discussion group where a number spoke of the importance of ACP forming part of an ongoing dialogue where they 'walked alongside' the person with MS and where ACP benefits derived more from its process than the written plan it produced, a sentiment echoed elsewhere in the recent literature [30]. However, this view was at odds among participants with MS where they believed their engagement in ACP was largely initiated privately within families, not helped by fragmented and incomplete information shared with them from services.

Additionally, once many participants transitioned from an active treatment pathway to a supportive pathway, appointments with health professionals become more sporadic. Typically, appointments focused on symptom management, referrals to other services and advice on other practical issues for example accessing benefits. ACP was never mentioned, and participants were sceptical if there would ever be time to initiate discussions. This might also explain why many participants in our study either had not engaged in ACP or were familiar with it as a concept. Our study participants also stated, that unlike seeing the same neurologist where continuity of care was similarly reflected as important in the ethical discussion group, they rarely benefited from an ongoing relationship with a single GP where ACP could be broached. Interestingly, this finding is also at odds with a recent observational cohort study that identified a key role in family doctors in initiating ACP-related conversations that resulted in

significantly fewer hospital deaths for those who made them [8]. Nevertheless, they recognised that if ACP were to be initiated by health professionals it needed to be tactfully and sensitively introduced by someone with appropriate skills.

Findings from the ethical discussion group shed light on the reality that some professionals, specifically MS clinical nurse specialists were fearful of engaging in ACP conversations and is even more important given that those with MS considered they might be the most appropriate professional to have these conversations with. This observation is evident in the wider literature, usually associated with a fear that patients may experience loss of hope [39, 40, 70, 74]. This may also be fuelled by health professionals who also fail to effectively discuss the reality of death and dying [40, 80–82] a situation widespread in post-modern Western culture [82, 83]. Nevertheless, communication is a fundamentally essential skill that can be learnt within the context of delivering ACP [84, 85] and has recently been encouraged by the European Academy of Neurology [35].

### Fear of being a burden

It is important to note many of our study participants' priorities and energy were focused on optimising current living, function, care, and quality of life rather than thinking about ACP. However, some feared being a burden to family members. This motivated them to engage in ACP-related processes, for example, LPA and ADRT. Other studies have similarly reported that documenting future wishes has the potential to reduce decisional burden, help caregivers avoid regret [74], constitutes a personal responsibility to family members [80], and act as a catalyst to ease tension in family negotiations [74]. Whilst these sentiments were present in our study, we observed some participants were concerned about their fate, should they outlive their family members. It is also worth noting that two participants who made DNACPR orders subsequently withdrew them after families expressed distress that they would refuse life-sustaining treatment, particularly if there was a chance their life could be saved.

### Enabling control and autonomy

It has been previously been acknowledged that people living with MS struggle to maintain a feeling of control in the face of an unpredictable disease course and uncertain future [86]. Intuitively, therefore ACP might extend zones of personal autonomy and involvement in decision-making beyond the stage where an individual loses their ability to make decisions or where communicating wishes may be lost [87]. In several cases, we observed LPAs represented a tool to enable control and autonomy. However, making an advance statement was viewed as relevant only if it drew health professionals' attention to their wishes and preferences. Our participants were sceptical that this would be the case, particularly as they were not officially recorded on health systems for others to view.

### Previous experiences of witnessing poor deaths

ACP has been reported as a means of mitigating the fear of experiencing a distressing or 'bad' death [82, 88]. In this study, previous experiences of witnessing dying were a clear motivator for some participants to engage in ACP, particularly the decision to make LPA. However, having MS and the prospect of dying were not viewed as the main reasons for engaging in ACP. Indeed, several participants stated explicitly they did not feel they were dying from their condition. For these participants, there was a fundamental belief that ACP and shared-decisional aids were relevant to them in the same way that they might be to anyone else, regardless of health status or condition

## Conclusions and clinical and research implications

In our study, people with MS and their families identified or asserted the importance of appropriate, accessible and meaningful information and dialogue to answer questions and uncertainties that would enable and empower them to be able to consider their future. However, this was often absent, a finding present in other studies involving people with MS [89, 90]. Our study participants described a sense of being alone in learning about and managing transitions of their illness. This was particularly so at the 'junction' of disease-modifying therapy to that of supportive care and set within in a context of fragmented or absent information. We believe this is compounded by health professionals, as evidenced in our study, not having the space, training or confidence to sensitively and adequately explore these issues with people with MS or their families, especially in the context of a disease in which some people will die [91]. Troublingly, participants' accounts would suggest this was present among those working in palliative care. Whilst the focus of many health professionals involved in MS in the UK and elsewhere has, appropriately, been one of actively living with MS, this may have then marginalised those wishing to explore end-of-life issues and ACP. And yet a third of participants had independently made plans, typically LPAs. The question, however, given this disconnect with these plans and health services, is whether they are the most appropriate solutions for these individuals, or if they are enacted to the best effect. We have shown that in previous recent research, where health professionals are involved and this information is communicated to services, preferences can be successfully realised [8].

Implementing high-quality ACP in all its guises relevant to those with MS requires an understanding of how ACP conversations and specific strategies can be brought to occur in a systematic, skilled, empathetic, and consistent manner within and across services. This demands the attention of multiple considerations that reinforce implementing high-quality, holistic, multi-component, and person/family-centric ACP relevant to MS. Bradshaw and colleagues believe a way of operationalising this is best done through a socioecological lens whereby a 'whole systems strategic approach' is employed [92, 93] and acknowledges that multiple, interconnected components must be present for this to be successful [94]. For this study, Fig 1 presents a model of how this approach attends to the individual, interpersonal, organisational, systems and time-based considerations present in our findings. Among other issues, they include the person with MS being knowledgeable about MS illness progression and how that might influence their decision-making, adequate training being available for health professionals about how to engage in ACP-related conversations, efficient IT and administration systems to share patient-based preferences and values, and seeing ACP as a process over time, not a one-time event and revisiting/repeating conversations where necessary as MS progresses.

Another pressing issue evident in this study is the requirement inherent with ACP for people, principally but not exclusively, living with life-limiting illnesses is to picture choices they would make when facing an uncertain future or health crisis. This is complicated by not knowing how they would feel or what would be important to them in those moments, a situation compounded by incomplete information [22]. MS is typically a progressive condition, punctuated by potentially life-changing and or life-threatening events, where shifts during this condition require people with MS and health professionals to be willing, flexible and prepared to rapidly manoeuvre when prognostic or other disease-related news is surprising, 'bad' or overwhelming. It has recently been suggested the phrase 'advance care planning' should be refashioned to 'adaptive care planning' to reflect the dynamic nature of decision-making that occurs along the continuum of serious or life-threatening illness [95]. People with MS can consequently make decisions that are more responsive or adaptive to the situation that reflects their potentially life-changing situations, for example, the need for assistive aids in the context of

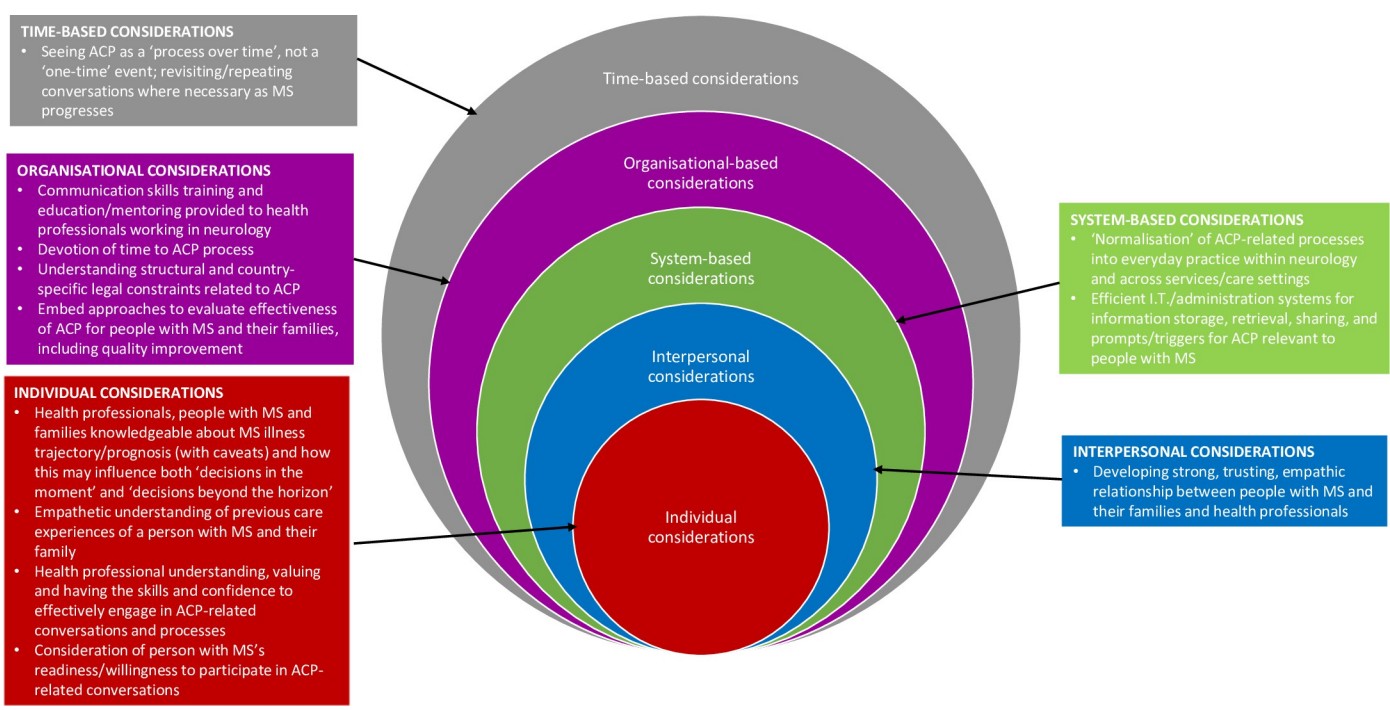

**Fig 1. Multi level considerations for advance care planning relevant to people with MS and their families.**

their illness as it unfolds. Further development on this approach would include a hybrid version that embraces ACP as a multi-component process but counteracts the false dichotomy of viewing ACP as either decisions 'made in the moment' or for decisions just 'beyond the horizon'. Both are relevant in MS and closely align with the recent UK General Medical Council's Guidance on Decision Making and Consent [96]. One virtue of allowing the opportunity for planning for the future as it unfolds or just beyond the horizon is that it permits health professionals to preserve the trust of people with MS and their families, titrate medical information concerning their deteriorating abilities, regularly review parallel care plans in which two sets of ACP are made; one for stability or improvement, and another for potential further deterioration, highly likely in the severe stages of MS [97]. Participants in the ethical discussion group spoke of this occurring in their practice but it is possible these discussions were too nuanced for patients to be aware of. The union of these types of ACP may enhance one another, allowing person/family-centric communication in ways that prepare all involved for making demanding decisions soon, whilst maintaining flexibility for adaptive and reactive decisions to be made 'in the moment'.

We suggest future research take place that informs the quality improvement associated with the implementation and delivery of ACP among people living with MS and their families who wish to engage in this activity. It has been suggested that the 'gold-standard' randomised controlled trial design widely used in ACP does not adequately address the 'context-specific drivers' behind implementation outcomes and their relationship to the underlying theory [22]. We therefore suggest researchers consider using a realist approach alongside traditional designs like the hybrid trial approach that include multiple methods. Realist evaluation is increasingly being employed to examine complex healthcare interventions, for example, ACP, as it seeks to provide a more in-depth understanding of what works, for whom and in what circumstances [98, 99].

## Supporting information

**S1 Appendix. Topic guide for people living with multiple sclerosis.**
(DOCX)

**S2 Appendix. Topic guide for ethical discussion group.**
(DOCX)

## Acknowledgments

We wish to extend our profound thanks to all those people with MS and their families and health professionals who participated in the interviews and the ethical discussion group. We thank members of the MS Society Research Network specifically, Alison Mack, Julie Clear, Mary Douglas, Patrick Burke and Rebecca Perry for sharing their time in the design of this study.

**Disclaimer:** The views expressed in this publication are those of the author(s) and not necessarily those of the NHS, NIHR, the Department of Health or the MS Society.

## Author Contributions

**Conceptualization:** Jonathan Koffman, Bobbie Farsides, Richard Nicholas, Eli Silber.

**Data curation:** Clarissa Penfold, Laura Cottrell, Bobbie Farsides.

**Formal analysis:** Jonathan Koffman, Clarissa Penfold.

**Funding acquisition:** Jonathan Koffman, Bobbie Farsides, Rachel Burman, Richard Nicholas, Stephen Ashford, Eli Silber.

**Investigation:** Jonathan Koffman, Clarissa Penfold, Laura Cottrell, Bobbie Farsides.

**Methodology:** Jonathan Koffman, Clarissa Penfold, Bobbie Farsides, Eli Silber.

**Project administration:** Jonathan Koffman, Clarissa Penfold, Laura Cottrell.

**Supervision:** Jonathan Koffman.

**Validation:** Jonathan Koffman, Clarissa Penfold, Bobbie Farsides.

**Visualization:** Jonathan Koffman.

**Writing – original draft:** Jonathan Koffman, Clarissa Penfold.

**Writing – review & editing:** Jonathan Koffman, Clarissa Penfold, Laura Cottrell, Bobbie Farsides, Catherine J. Evans, Rachel Burman, Richard Nicholas, Stephen Ashford.

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
