## [Decision Letter · Decision Letter 0]

7 Apr 2022

PONE-D-22-06947“I wanna live and not think about the future” What place for advance care planning for people living with severe multiple sclerosis and their families? A qualitative studyPLOS ONE Dear Dr. Koffman

Thank you for submitting your manuscript to PLOS ONE. After careful consideration, we feel that it has merit but does not fully meet PLOS ONE’s publication criteria as it currently stands. Therefore, we invite you to submit a revised version of the manuscript that addresses the points raised during the review process.

We look forward to receiving your revised manuscript.

Kind regards,

Luigi Lavorgna

Academic Editor

PLOS ONE

Journal Requirements:

5. Thank you for stating the following in the Funding Section of your manuscript: 

"This study was generously funded by the MS Society [Grant code 93]. Catherine Evans is funded by a Health Education England/ NIHR Senior Clinical Lectureship (ICA-SCL-2015-01-001). The views expressed in this publication are those of the authors and not necessarily those of the NHS, the National Institute of Health Research or the Department of Health and Social Care."

"This study was generously funded by the MS Society [Grant code 93]. Catherine Evans is funded by a Health Education England/ NIHR Senior Clinical Lectureship (ICA-SCL-2015-01-001). The views expressed in this publication are those of the authors and not necessarily those of the NHS, the National Institute of Health Research or the Department of Health and Social Care."

7. Please include your tables as part of your main manuscript and remove the individual files. Please note that supplementary tables (should remain/ be uploaded) as separate "supporting information" files

Reviewers' comments:

Reviewer's Responses to Questions

**Comments to the Author**

1. Is the manuscript technically sound, and do the data support the conclusions?

Reviewer #1: Yes

2. Has the statistical analysis been performed appropriately and rigorously? 

Reviewer #1: Yes

3. Have the authors made all data underlying the findings in their manuscript fully available?

Reviewer #1: Yes

4. Is the manuscript presented in an intelligible fashion and written in standard English?

Reviewer #1: Yes

5. Review Comments to the Author

Reviewer #1: Very interesting and well developed work. I should have some remarks:

1)People with MS have to learn facing everyday situation and also think to the future situation. Deciding to accepet ACP could be a way to cope the uncertainty of the future and being seen as a coping strategy ( Santangelo, G., Corte, M. D., Sparaco, M., Miele, G., Garramone, F., Cropano, M., Esposito, S., Lavorgna, L., Gallo, A., Tedeschi, G., & Bonavita, S. (2021). Coping strategies in relapsing-remitting multiple sclerosis non-depressed patients and their associations with disease activity. Acta neurologica Belgica, 121(2), 465–471. https://doi.org/10.1007/s13760-019-01212-5)

Please discuss it in the introduction.

2) It was not taken into consideration if the patients were religious, why? Could the authors this could add more data to what has already been shown? Sparaco, M., Miele, G., Abbadessa, G., Ippolito, D., Trojsi, F., Lavorgna, L., & Bonavita, S. (2021). Correction to: Pain, quality of life, and religiosity in people with multiple sclerosis. Neurological sciences : official journal of the Italian Neurological Society and of the Italian Society of Clinical Neurophysiology, 10.1007/s10072-021-05814-x. Advance online publication. https://doi.org/10.1007/s10072-021-05814-x

Discuss it.

6. PLOS authors have the option to publish the peer review history of their article (what does this mean?). If published, this will include your full peer review and any attached files.

Reviewer #1: No

---

## [Author Response · Author response to Decision Letter 0]

21 Apr 2022

21st April 2022

Dear Dr Lavorgna

I do hope this letter finds you well. Thank you very much for taking the time to read our manuscript entitled, “I wanna live and not think about the future” What place for advance care planning for people living with severe multiple sclerosis and their families? A qualitative study to be considered for publication in PLoS One. I have read your very helpful comments and those of Reviewer 1 in detail and have made the necessary changes as required. I outline them below

1. As you are aware our study represents a qualitative inquiry of the place and meanings of advance care planning for people living with MS and was not a laboratory-based study. We, therefore, do not have a ‘laboratory protocol’ to enhance the reproducibility of our results. 

2. I have reviewed all the references cited in our manuscript in detail to ensure that they are complete and correct. None of the publications cited has been retracted. I have made a slight change to the references cited to include the two very helpful publications suggested by Reviewer 1 (please refer to changes to the manuscript below that mention these two additional references).

3. I have now provided additional details regarding participant consent. In the ethics statement in the Methods and online submission information, I have specified the type of consent obtained for our study, in this case, ‘written informed consent’. This is noted on page 8 of the revised manuscript. As stipulated, we note that ethical approval for our study was provided in the Methods section of the manuscript. 

4. Thank you for pointing out that the grant information I provided in the ‘Funding Information’ and ‘Financial Disclosure’ sections do not match. I have now removed the financial disclosure information from the manuscript and inserted this information as required in the relevant section of the online submission of the manuscript as follows: 

5. As per point 4 when resubmitting the manuscript, I have now ensured that I provide the correct grant numbers for the awards we received for our study in the ‘Funding Information’ section. I hope this now manages all issues associated with the finance of the study, so you do not need to change the online submission form on my behalf.

6. Thank you for raising your concern about the Data Availability statement. I have now provided an answer to this section of the online submission that reads ‘All relevant data are within the manuscript and its Supporting Information files’. This statement complies exactly with the most relevant statement suggested by the online submission of the manuscript.

7. I have now included all the tables as part of the main manuscript and have removed all the individual files. I have, however, retained Figure 1 as a separate file that subsequently needs to be incorporated within the manuscript and indicate where it needs to reside. Please refer to page 41 of the revised manuscript. 

RESPONSE TO REVIEWER #1:

We thank reviewer 1 for stating our work was very interesting and well developed. Below I have responded to each of his/her comments 

1. People with MS have to learn facing everyday situation and also think to the future situation. Deciding to accept ACP could be a way to cope the uncertainty of the future and being seen as a coping strategy (Santangelo, G., Corte, M. D., Sparaco, M., Miele, G., Garramone, F., Cropano, M., Esposito, S., Lavorgna, L., Gallo, A., Tedeschi, G., & Bonavita, S. (2021). Coping strategies in relapsing-remitting multiple sclerosis non-depressed patients and their associations with disease activity. Acta neurologica Belgica, 121(2), 465–471.https://doi.org/10.1007/s13760-019-01212-5)

Please discuss it in the introduction.

Reviewer 1 raises an important issue. Rather than mentioning this issue in the introduction of the manuscript, I believe it fits better in the Discussion section of the revised manuscript. Specifically, on page 35 of the tracked changed revised manuscript I now write:

For some people with MS, punctuated losses associated with clinically significant disease events (41, 71-74) [41, 71-74] may be associated with positive coping strategies (6) leading to acceptance of MS as a progressive condition and the creation of a new self-identity where ACP may become relevant.

Reference 6 refers to the suggested publication:- Santangelo, G., Corte, M. D., Sparaco, M., Miele, G., Garramone, F., Cropano, M., Esposito, S., Lavorgna, L., Gallo, A., Tedeschi, G., & Bonavita, S. (2021). Coping strategies in relapsing-remitting multiple sclerosis non-depressed patients and their associations with disease activity. Acta neurologica Belgica, 121(2), 465–471. https://doi.org/10.1007/s13760-019-01212-5

In addition, I also state on page 39 of the revised tracked changed manuscript I state that:

Identifying which people living with MS are less able to develop strategies to cope with their situation especially when uncertainty is omnipresent (6), should lead health professionals to address their potentially modifiable state with psychological interventions, for example, cognitive behaviour therapy (76).

2. It was not taken into consideration if the patients were religious, why? Could the authors this could add more data to what has already been shown? Sparaco, M., Miele, G., Abbadessa, G., Ippolito, D., Trojsi, F., Lavorgna, L., & Bonavita, S. (2021). Correction to: Pain, quality of life, and religiosity in people with multiple sclerosis. Neurological sciences : official journal of the Italian Neurological Society and of the Italian Society of Clinical Neurophysiology, 10.1007/s10072-021-05814-x. Advance online publication. https://doi.org/10.1007/s10072-021-05814-x

Discuss it.

We thank Reviewer 1 for raising this important issue. On page 37 of the revised tracked changed manuscript in the ‘strengths and limitations’ section, I now state that whilst we did not explore study participants’ perspectives about religiosity and spirituality and previous research has observed that religiosity has an influence on the relevance and acceptability of advance care planning (63-64) which may also be present among those living with MS (65).

Reference 65 refers to the suggested publication: - Sparaco, M., Miele, G., Abbadessa, G., Ippolito, D., Trojsi, F., Lavorgna, L., & Bonavita, S. (2021). Correction to: Pain, quality of life, and religiosity in people with multiple sclerosis. Neurological sciences : official journal of the Italian Neurological Society and the Italian Society of Clinical Neurophysiology, 10.1007/s10072-021-05814-x. Advance online publication. https://doi.org/10.1007/s10072-021-05814-x

I have uploaded two versions of the reviewed manuscript, the first with tracked changes to highlight all the changes made that correspond with both your and Reviewer 1’s comments and the second without tracked changes.

Thank you ever so much for the opportunity to revise our manuscript which I hope is satisfactory. I look forward to hearing from you soon. 

All good wishes,

Jonathan Koffman

Professor in Palliative Care 

Email: jonathan.koffman@hyms.ac.uk

---

## [Editor Report · Decision Letter 1]

9 May 2022

“I wanna live and not think about the future” What place for advance care planning for people living with severe multiple sclerosis and their families? A qualitative study

PONE-D-22-06947R1

We’re pleased to inform you that your manuscript has been judged scientifically suitable for publication and will be formally accepted for publication once it meets all outstanding technical requirements.

Kind regards,

Luigi Lavorgna

Academic Editor

PLOS ONE
---

## [Editor Report · Acceptance letter]

11 May 2022

PONE-D-22-06947R1 

“I wanna live and not think about the future” What place for advance care planning for people living with severe multiple sclerosis and their families? A qualitative study  

Dear Dr. Koffman:

I'm pleased to inform you that your manuscript has been deemed suitable for publication in PLOS ONE. Congratulations! Your manuscript is now with our production department. 

Kind regards, 

on behalf of

Dr. Luigi Lavorgna 

Academic Editor

PLOS ONE